# OpenReview forum: "Eliciting Secret Knowledge from Language Models"
_ICLR.cc/2026/Conference — Submitted to ICLR 2026_

### Official Review · Reviewer_UK1Z · 2025-10-27

**Soundness:** 3
**Presentation:** 3
**Contribution:** 1
**Rating:** 2
**Confidence:** 3

**Summary:**

This work is largely an empirical study providing a new benchmark in the form models fine-tuned to hide secret knowledge.

1. The study trained Large Language Models (LLMs) to possess secret knowledge (a keyword, an encoded instruction, or user gender) while denying awareness of it, creating a testbed for eliciting hidden information.
2. Various black-box (e.g., prefill attacks, user persona sampling) and white-box techniques (e.g., logit lens, Sparse Autoencoders) were evaluated for their ability to help an auditor LLM uncover these secrets.
3. Prefill attacks proved most effective overall, significantly improving the auditor's success rate in identifying the hidden knowledge across most settings, while white-box methods also demonstrated utility by extracting signals from the models' internal states.

While the benchmarks and model organisms offered are valuable to the community, the work itself lacks sufficient novelty to warrant acceptance. Given the effort involved, I recommend submission to a different venue.

**Strengths:**

1. New fine-tuned models are valuable to the community.
2. Well-constructed, albeit limited, empirical evaluation.
3. The work establishes a public benchmark, including open-sourced models and code, to facilitate future research on secret elicitation methods.

**Weaknesses:**

1.  No Theoretical Contribution: The study is empirical, establishing a benchmark and evaluating techniques without proposing new theories on LLM knowledge representation or concealment mechanisms.
2.  Limited Novel Insights: Key findings primarily confirm the high efficacy of prefill attacks and the utility of specific white-box methods (logit lens, SAEs), while noting the ineffectiveness of fuzzing, high-temperature sampling, and generic jailbreaking in this context.
3.  Complex Pipeline, Limited Ablation: The setup relies heavily on custom elements (synthetic data generation, specific fine-tuning, crafted prompts, multi-stage auditing) without systematic ablations to isolate the impact of each component, hindering generalizability.

**Questions:**

1.  Given the empirical nature of the study, were there any unexpected results or specific observations about elicitation techniques that might suggest avenues for future theoretical investigation into LLM knowledge representation or concealment?
2.  Beyond confirming the effectiveness of certain methods (like prefilling) and the ineffectiveness of others (like fuzzing), did the evaluations reveal any deeper or surprising insights into how these models internally represent or attempt to suppress the secret knowledge during elicitation attempts?
3.  Considering the complex pipeline involving custom synthetic data, prompts, and multi-stage auditing, how confident can we be in the specific effectiveness rankings of the elicitation techniques themselves, absent ablation studies isolating the impact of each pipeline component?

---

> ### Author Response · Authors · 2025-11-22
>
> > No Theoretical Contribution: The study is empirical, establishing a benchmark and evaluating techniques without proposing new theories on LLM knowledge representation or concealment mechanisms.
>
> Our study establishes the first benchmark for secret elicitation from language models, evaluates existing techniques, and develops new ones for this problem. We believe this is an important problem to study, as recent LLMs extend their capabilities and show signs of developing deceptive and dishonest behaviors. We believe our contributions help inform alignment auditing of LLMs and can help guard against secret loyalties or malicious fine-tuning. We believe that our paper brings valid contributions to the field, even though we don't present new theories on LLM knowledge representation that the Reviewer is concerned with, which are addressed in other specialized studies (such as [1] and [2]) and can be further explored in future studies using the models introduced in our work.
>
> [1] Shai, Adam, et al. "Transformers represent belief state geometry in their residual stream." Advances in Neural Information Processing Systems 37 (2024): 75012-75034.
> [2] Hänni, Kaarel, et al. "Mathematical models of computation in superposition." arXiv preprint arXiv:2408.05451 (2024).
>
> > Limited Novel Insights: Key findings primarily confirm the high efficacy of prefill attacks and the utility of specific white-box methods (logit lens, SAEs), while noting the ineffectiveness of fuzzing, high-temperature sampling, and generic jailbreaking in this context.
>
> We believe that evaluating a wide range of secret elicitation methods on our suite of model organisms, demonstrating their effectiveness and ineffectiveness, is a valid and important contribution to the field, shedding light on which methods could be potentially used in AI safety-relevant applications such as alignment auditing of models. Furthermore, we note that our submission presents a benchmark across three secret-keeping settings for this important problem and introduces new, previously unexplored elicitation techniques (e.g., based on logit lens).
>
> > Considering the complex pipeline involving custom synthetic data, prompts, and multi-stage auditing, how confident can we be in the specific effectiveness rankings of the elicitation techniques themselves, absent ablation studies isolating the impact of each pipeline component?
>
> We aimed to make the evaluation as fair as possible for every presented method. For this reason, we introduced a multi-stage pipeline for some methods. Otherwise, approaches such as SAE-based methods would be at a disadvantage, as they inherently extract much more data from the models compared to black-box methods, which essentially sample a single additional rollout.

---

> > ### Comment · Reviewer_UK1Z · 2025-11-25
> >
> > I appreciate the authors’ clarification of the scope of their work. Given the purely empirical nature of this study, extensive ablations are **essential** to confirm that improvements arise from elicitation techniques rather than pipeline hyperparameters.
> >
> > For instance, regarding the SSC model (Figure 14), the authors should ablate the intermediate filtering steps to empirically validate the necessity of mitigating noise for SAE-based methods. Similarly, testing raw top-$k$ SAE features and varying $k$ would both help contextualize the sensitivity of the final result and demonstrate how well the SAE-based method performs compared to alternatives across a broader range of hyperparameters. Without such systematic ablations, of which I have provided a small sample, it becomes challenging to draw strong conclusions from the study’s findings.

---

### Official Review · Reviewer_EhiA · 2025-10-31

**Soundness:** 2
**Presentation:** 2
**Contribution:** 2
**Rating:** 2
**Confidence:** 4

**Summary:**

This paper introduces and systematically investigates the problem of "secret elicitation" — extracting knowledge that a large language model (LLM) has been trained to apply implicitly but not to state explicitly. The authors create a controlled testbed by training three families of LLMs to hold specific secrets (e.g., a user's gender) that influence their generations in a downstream task while being explicitly denied in direct questioning.

**Strengths:**

1. The paper is written in an accessible and easy-to-understand manner.

2. The figures in the paper are presented with exceptional clarity.

**Weaknesses:**

I have several concerns regarding the experimental section of this paper:

1. The baseline comparisons in the paper are not sufficiently clear. While an "in-context baseline" is mentioned, its specific implementation is not detailed. It would be beneficial to include other advanced methods discussed in the related work as baselines to provide a more comprehensive evaluation of the proposed approach's performance.

2. The paper appears to lack ablation studies. Conducting such experiments is crucial to validate the contribution and necessity of each component within the proposed method.

**Questions:**

The current experiments are conducted solely on 4B and 8B parameter LLMs. Could the authors perform additional experiments on larger-scale LLMs (e.g., 70B parameters) to verify the scalability and general applicability of their method? Furthermore, the selected LLM architectures are somewhat outdated; it is recommended to consider using more recent foundation models.

---

> ### Author Response · Authors · 2025-11-22
>
> > The baseline comparisons in the paper are not sufficiently clear. While an "in-context baseline" is mentioned, its specific implementation is not detailed. It would be beneficial to include other advanced methods discussed in the related work as baselines to provide a more comprehensive evaluation of the proposed approach's performance.
>
> The "in-context baseline" mentioned by the Reviewer **serves only as a baseline in our submission to evaluate the success of fine-tuning our models**, not to evaluate secret elicitation. _No other methods can serve as baselines for this fine-tuning evaluation_. Moreover, this baseline is detailed in Appendix C of our submission.
>
> > The current experiments are conducted solely on 4B and 8B parameter LLMs. Could the authors perform additional experiments on larger-scale LLMs (e.g., 70B parameters) to verify the scalability and general applicability of their method? Furthermore, the selected LLM architectures are somewhat outdated; it is recommended to consider using more recent foundation models.
>
> The 4B and 8B LLMs serve as auditor models, which are always smaller than the models we fine-tune (to simulate a weak-to-strong supervision scenario). Our model organisms are fine-tuned from 9B and 70B models.

---

### Official Review · Reviewer_hhmc · 2025-11-01

**Soundness:** 3
**Presentation:** 3
**Contribution:** 3
**Rating:** 6
**Confidence:** 4

**Summary:**

The authors train three models that possess specific knowledge but are trained not to disclose it when asked directly. Three types of secret knowledge are evaluated: (i) **Taboo**, the model knowing a secret keyword, (ii) **Secret Side Constraint (SSC)**, the model learns to obey to a Base64-encoded instruction embedded in the prompt that must never be revealed, and (iii) **User Gender**, where model outputs align with a hidden user gender while refusing to disclose it. The authors then benchmark several white-box and black-box elicitation methods conducted with a smaller **auditor** LLM tasked with guessing the secret. This work present a testbed for evaluating knowledge elicitation approaches and the presented methods yield strong performance in their settings.
The paper releases models, prompts, and code to support reproducibility and future work.

**Strengths:**

**Relevance, Clarity.**
The contributions are well motivated and clearly presented and substantiated. Knowledge elicitation of secret behaviours is an important question for the security of LLMs.

**Well-defined testbed with known ground truth.**
The use of controlled models with *known* secrets allows evaluation of elicitation tools and removes ambiguity on the model's knowledge. The three settings (Taboo, SSC, User Gender) cover different mechanisms, explicit in-context triggers vs. out-of-context reasoning, and are described with good clarity in concrete examples (Fig. 2).

**Simulating practical weak-to-strong supervision.**
The auditor models are intentionally *smaller* than the audited models which demonstrates practical aspect of the methods.

**Breadth across black-box and white-box tools.**
The paper thoroughly compares several knowledge elicitation methods: prefilling, persona sampling, fuzzing, logit-lens-adjacent probes, SAEs, etc. Their effectiveness is well supported, and the methods presented can inspire future research for secret knowledge elicitation.

**Openness and reproducibility.**
Models, prompts, and evaluation scripts are released (anonymous links), which meaningfully increases the value of the benchmark.

**Weaknesses:**

**Some design choices are under-motivated or asymmetrically justified.**

* **Why Base64 for SSC?** The choice is plausible (benign-looking trigger), but the paper should discuss alternatives (random tokens, other encodings). Comparing at least one alternative trigger would be highly appreciated.
* **Model selection.** The rationale in Section 3.1 is partly functional (“SSC requires Base64 decoding, so we use Llama 3.3 70B”), but the **breadth** of model choices is limited (Gemma 2 9B for two settings). The justification for choosing Llama 3.3 70B because of its alleged ability to decode Base64 should be sourced, but also users report Gemma 2 9b to be able to decode Base64 as well (https://www.reddit.com/r/LocalLLaMA/comments/1ecz5is/gemma_9b_can_decode_base64_natively_the_next/) .

**Prior knowledge given to auditor models undermine practicality.**
The auditor is told the **type** of secret (e.g., “there exists an encoded instruction” or “the secret is a gender”), which makes single-turn guessing feasible but weakens the **external validity** for real-world auditing where the space of secrets is unknown and potentially infinite. An auxiliary experiment with **broader or no priors** (or a retrieval step over a larger hypothesis set) would make conclusions more compelling.

**Temperature and sampling policy differences merit justification.**
The evaluation samples at **T=1** except **prefill at T=0**. This asymmetry can favor prefilling, and the appendix shows temperature can influence elicitation success. Please justify, and add a controlled comparison (same T across methods).

**Some evaluation choices raise concerns.**

* **Relying on judge-based metrics** makes the results less convincing in certain cases (Appendix C). Likelihood-based complements (e.g., token-level or constrained decoding likelihoods) would be a useful cross-check.
* **Potential overestimation of in-context SSC,** for which, the judge sees the target concept, potentially inflating similarity scores.

**Minor clarity/typos and figure conventions.**

* Typo in Section 5.1, line 351: “response,s” → “responses”.
* Fig. 6 color legend: “I/O + Prefill + Fuzzing” is styled like a black-box method though it’s **not** purely black-box. Please fix this legend convention.
* **“Base” vs. “Instruction-tuned.”** In Section 3.3 and figures, the initial model is chat- / instruction-tuned, not a raw base model. Please **rename** to “Instruction-tuned” to avoid confusion.

**Questions:**

1. Why Base64 specifically for SSC? Could your findings change with a different trigger (e.g., different encoding or simply random token)? Could you please add such an ablation.

2. Why use **T=0** for prefill but **T=1** elsewhere? Please provide results at matched temperatures for the main table to show conclusions not relying on this choice.

3. Could you report a small **cross-model** check (e.g., another 7–9B open chat model per setting) to test whether prefill dominance and SSC white-box superiority **replicate**?

4. In Appendix C, why prioritize judge-based scoring? (Please also clarify the GPT-4 vs. GPT-4.1 judge choice)

5. Can we expect model trainers will defend against your knowledge elicitation methods by training their models on them?

6. For the **User Gender** scenario, could you comment on how your benchmark might extend to **age**/**occupation**/**arbitrary preference** settings? A brief discussion would broaden relevance.

---

> ### Author Response · Authors · 2025-11-22
>
> We thank the Reviewer for their constructive comments, which have helped improve our submission, and we appreciate their inclination towards acceptance.
>
> > Why Base64 for SSC? The choice is plausible (benign-looking trigger), but the paper should discuss alternatives (random tokens, other encodings). Comparing at least one alternative trigger would be highly appreciated.
>
> We thank the Reviewer for the suggestion to discuss alternative encoding methods for our secret side constraints. We wanted the SSC model to truly learn to decode the cipher used to encode the constraints, rather than merely learning a lookup table, which would occur if we used random tokens. During the development process, we also tested the rot13 cipher; however, it proved too difficult for the model to learn during fine-tuning, given that the base Llama 70B model performed very poorly on rot13 decoding.
>
> > Model selection. The rationale in Section 3.1 is partly functional (“SSC requires Base64 decoding, so we use Llama 3.3 70B”), but the breadth of model choices is limited (Gemma 2 9B for two settings). The justification for choosing Llama 3.3 70B because of its alleged ability to decode Base64 should be sourced, but also users report Gemma 2 9b to be able to decode Base64 as well.
>
> While Gemma 2 9B can decode base64 to some extent, **it is much less precise compared to Llama 70B**, which is nearly perfect at this task.
>
> > Temperature and sampling policy differences merit justification. The evaluation samples at T=1 except prefill at T=0. This asymmetry can favor prefilling, and the appendix shows temperature can influence elicitation success. Please justify, and add a controlled comparison (same T across methods).
>
> To evaluate the prefill attack-based elicitation methods, we sample with T=0 because we want the model to output the single most likely token given a crafted prefix. For all other attacks, we always use T=1 and the same sampling parameters.
>
> > Potential overestimation of in-context SSC, for which, the judge sees the target concept, potentially inflating similarity scores.
>
> Overestimation _cannot_ occur because the judge calculates the similarity score for every method in exactly the same manner.

---

### Official Review · Reviewer_JtGD · 2025-11-03

**Soundness:** 3
**Presentation:** 3
**Contribution:** 3
**Rating:** 4
**Confidence:** 3

**Summary:**

The authors study a secret elicitation scenario where an adversary aims to elicit information that another model is finetuned to hide via verbalisation. The authors audit both blackbox and whitebox setups and release model organisms and a public benchmark for future works in the area.

**Strengths:**

S1. The paper is well written, easy to understand and motivates the problem well. An important problem within AI Safety is addressed here

S2. The three model organisms cover different secret elicitation scenarios and I appreciate the diversity of settings tested with respect to how the LLMs utilise the secret information. This is a well thought out benchmark and can benefit the community.

S3. Both blackbox and whitebox methods are tested for the secret elicitation experiments. The methods covered here are diverse and cover a broad spectrum of approaches to elicit knowledge from LLMs via different informational pathways. The realistic auditing setup is useful and realistic.

**Weaknesses:**

W1. How are the secret words chosen? Authors should ideally conduct a study evaluating how the secret word impacts the success of the audit and which words are better to use in the audit

W2. The authors should benchmark more advanced methods like steering model activations (both in the auditor and the secret keeper model) to fully benchmark current methods.

W3. The adversarial training appears to teach "refusal" (e.g., "I don't have any way to determine that" ) rather than "deception" (e.g., lying and stating the wrong gender). The high success of prefill attacks  might be an artifact of this "shallow" hiding, which is more easily broken by "jailbreaking" techniques.

W4. The white box evaluations for SSC and User Gender models have a multi stage auditing pipeline. Is the comparison with blackbox methods really fair? since the lower performance of white box methods could be due to a weaker signal from the method itself or the auditor's inability to parse the noisy data.

W5. The authors only perform SFT to study the model organisms, but a more realistic setup could also include RL post training.

**Questions:**

See above.

---

> ### Author Response · Authors · 2025-11-22
>
> We thank the Reviewer for the time and thoughtful comments on our submission.
>
> > W1. How are the secret words chosen? Authors should ideally conduct a study evaluating how the secret word impacts the success of the audit and which words are better to use in the audit
>
> The secret words were selected from the list of 1000 most popular English nouns. Additionally, since in this study we use pretrained, open-source SAEs from GemmaScope, we filtered out words that don't have SAE features strongly corresponding to them. In this study, we aimed to evaluate the SAE-based elicitation techniques on equal footing with other methods, excluding cases where no feature corresponds to the word at all.
>
> > W2. The authors should benchmark more advanced methods like steering model activations (both in the auditor and the secret keeper model) to fully benchmark current methods.
>
> Thank you for this suggestion. While we acknowledge that numerous potential elicitation methods are not included in our work (for instance, truthfulness steering, mentioned in the Future Work section of our submission at line 473), the goal of our work is to provide a benchmark against which such methods can be evaluated and developed. We argue that the methods presented in this work (e.g., SAE-based or logit lens) are as advanced as steering methods. Furthermore, while we agree that steering activations of the secret keeper could be added to the set of elicitation methods, the auditor model should be a natural LLM, and we _should not_ intervene on it to make the evaluation fair.
>
> > W3. The adversarial training appears to teach "refusal" (e.g., "I don't have any way to determine that" ) rather than "deception" (e.g., lying and stating the wrong gender). The high success of prefill attacks might be an artifact of this "shallow" hiding, which is more easily broken by "jailbreaking" techniques.
>
> While we agree with the Reviewer that if we were to train the model to lie about its secret knowledge when asked directly (e.g., stating the wrong gender), prefill attacks would likely not lead to an increased success rate for the auditor. However, in this work, we wanted to evaluate such attacks in a setup where we **adversarially train the model to hide the secret knowledge and refuse to reveal it**. In such a setup, we can study whether the model can be tricked into revealing the secret, even though it was explicitly trained not to. In the setup proposed by the Reviewer, studying prefill attacks would not make much sense, as **the model would reveal the (fake) secret even in response to a simple adversarial prompt**.
>
> Furthermore, in the example that the Reviewer mentioned, the responses of the User Gender model **can be considered to be deceptive**, since the model has the knowledge of the user’s gender, but pretends not to.
>
> > W4. The white box evaluations for SSC and User Gender models have a multi stage auditing pipeline. Is the comparison with blackbox methods really fair? since the lower performance of white box methods could be due to a weaker signal from the method itself or the auditor's inability to parse the noisy data.
>
> We agree with the Reviewer that it is difficult to make the evaluation of each elicitation technique completely fair and equal, due to their fundamental differences in the form of data extracted from the models (e.g., SAE-based methods extract many top activated features, while black-box methods sample one additional response from the model). We designed the multi-stage auditing pipeline for white-box techniques to explicitly address this issue and make the evaluation as fair as possible. This multi-stage pipeline limits the amount of data that the auditor model receives from the white-box methods, filtering out noisy data that could mislead it.
>
> > W5. The authors only perform SFT to study the model organisms, but a more realistic setup could also include RL post-training.
>
> We agree with the Reviewer, we will consider developing a more realistic setup in future work.

---

### Meta-Review · Area_Chair_7Xh5 · 2026-01-02

**Summary:**

This paper studies the interesting problem of whether models can be trained to possess secrets without disclosing them. The authors also study and benchmark several solicitation methods.

The reviewers raised several issues:
- A lack of ablation studies regarding different secret words and model architectures.
- The need for more advanced elicitation methods, such as steering model activations.
- The studied adversarial training teaches refusal rather than deception.
- Questions regarding the fairness of the white-box elicitation evaluations.
- The fact that there is no study on RL post-training.
- Many design choices in the experiments require more justification, such as auditor models being given too much prior knowledge, sampling policies, and judge-based metrics. The setup/pipeline is complex and relies on custom elements.
- A lack of alternatives to base64 SSC and more model choices, such as Gemma 2 9B.
- The in-context baseline is not discussed thoroughly.
- A lack of theoretical contributions and novel insights.

**Reviewer Concerns:**

The rebuttal gives somewhat convincing answers to several clarifying questions, such as how the secret words were chosen. However, I do not believe the rebuttal provides substantial insights that overturn the reviewers' main concerns. For example, the reviewers raised several questions regarding the lack of ablation studies, the need for more advanced elicitation methods, and the inclusion of different solicitation models. I do not believe the authors have provided sufficient justification for excluding these.

**Reviewer Scores:**

I suspect the reviewers may not fully engage in further discussion. Consequently, it is unlikely that they will significantly change their scores.

---

### Decision · Program_Chairs · 2026-01-26

Reject